# The Influence of Faith and Religion on Family Interactions and Interest in Health Issues during the COVID-19 Pandemic—A Study among Polish Adolescents

**DOI:** 10.3390/ijerph19116462

**Published:** 2022-05-26

**Authors:** Anna Kasielska-Trojan, Julian Dzierżak, Bogusław Antoszewski

**Affiliations:** 1Plastic, Reconstructive and Aesthetic Surgery Clinic, Medical University of Lodz, Kopcinskiego 22, 90-153 Lodz, Poland; boguslaw.antoszewski@umed.lodz.pl; 2Salesian School Complex, 90-046 Lodz, Poland; julian.dzierzak@wodna.edu.pl

**Keywords:** religiosity, pandemic, catholic school, COVID-19

## Abstract

Background: COVID-19 has dominated health, economic, and geopolitical issues for many months, but it also has great influence on individuals and families. The aim of this study was to verify whether the pandemic of COVID-19 changed religious practices and how religiosity moderated the influence of the pandemic on family interactions and attitudes towards health issues in adolescents. Methods: The study groups included 561 adolescent high school students (314 females and 247 males) from two kinds of high schools: public and Catholic. Results: Most Catholic school students have not changed their religious practices during the pandemic or just changed the form of attendance to TV or internet (59.7%). Moreover, 8.6% of them stopped the practices, in comparison with 12.9% of public school students, most of whom had not attended a mass before and during the pandemic. The results showed that in adolescents’ opinions the pandemic caused family relations to be stronger, however this effect was modified by religiosity. Conclusion: Attending Catholic school and being a practicing believer influenced some aspects of faith and family relations during the pandemic, but in most aspects, they did not influence attitudes toward health issues. The results of the study highlight the need to secure, especially for non-believing adolescents, family support during the pandemic, while in believers faith may provide such support.

## 1. Introduction

COVID-19 dominated health, economic, and geopolitical multipolar issues for many months (if not years) [1]. Recent research has revealed that fear of COVID-19 affects mental health, well-being, and behavior in adolescents who are, in this context, an especially vulnerable group [2,3,4,5]. The problem of the pandemic’s influence on different aspects of family life, social functioning, and general well-being among adolescents has been a subject of numerous studies. These studies provided different results. An Italian study concluded that on average, healthy parents and adolescents seemed to deal fairly well with the circumstances and that whether or not parents and adolescents experienced (emotional) problems could vary from household to household [6]. Many studies showed that the pandemic impacted families at different levels, such as changed family routines and rules, increased chaos at the family level, positive and negative changes in parent–child relationships, changes in the well-being of family members, and externalizing problems in children and adolescents [7,8,9,10,11]. Some researchers pointed toward positive effects on family dynamics in some families during the pandemic [10,12]. These positive perspectives involved cases where parents spent more time in activities with their children, compared with the pre-lockdown period [13]. Moreover, Günther-Bel et al. (2020) found that parents reported more relational improvement than deterioration during the lockdown period [14].

Additionally, high levels of fear and anxiety also affected religious beliefs and behaviors [15]. Kowalczyk et al. (2020) stated that a new “generation of coronavirus” are often subjected to “spiritual renewal” [16]. On the other hand, it was also shown that changes in religiosity during the COVID-19 pandemic would not always cause increased participation in rituals as it could make it easier for some to cease their religious practices [17].

Poland is regarded as a Catholic country of approximately 87% (2021), with 94% (1992) declaring themselves as ‘believers’ and approximately 76% of society declaring themselves as “regularly or irregularly practicing believers” (2021) (regardless of age, over 18 years old), but with approximately 36% “non-practicing” in the 18–24 years age group (data from the Centre for Public Opinion Research, 2021). These statistics indicate that the religiosity of adolescents and young adults is becoming weaker and there are more and more non-believers; however, the current changes in the religiosity of Poles should be interpreted in the wider aspects, and it should combine different perspectives as, on the other hand, the percentage of “strong believers” who practice several times a week remains stable or even increases. This indicates two different directions: secularization, atheization, and anticlericalism; and, contrarily, a stronger and a more conscious and consistent religiosity [18]. In this aspect, religiosity seems to be an important factor in considering the pandemic’s influence on different aspects of life, but also in presenting different attitudes to health issues. The study by Krok et al. (2021) concerning this aspect showed that meaning-making and fear of COVID-19 influenced the relationship between religiosity and well-being among late adolescents. The authors highlighted the indirect effects of meaning-making and fear of COVID-19 that revealed an interplay of cognitive (i.e., beliefs and goals) and emotional (i.e., fear and anxiety) processes in shaping adolescent well-being. They concluded that “young people who showed both higher levels of insight into themselves and their worldviews as a result of meaning-making and awareness of threats posed by COVID-19 were able to productively use their religious resources to maintain well-being” [19]. Similarly, it was reported that in Muslims choosing to “trust in God”, the faith can positively impact their levels of psychological resilience [20].

It thus seems to be important to examine how COVID-19 affected family interactions and how this reaction was modulated by religiosity among late adolescents. Although some studies have explored the potential effect of faith and religion on the positive and/or negative influence of the pandemic on family interactions, there is no research concerning their effect on attitudes to family health issues among Polish adolescents during the COVID-19 pandemic. Considering the school system in Poland, it is possible to reach such a target group in high-schools. The school system in Poland involves pre-school education (3–6 years), 8-year primary schools and secondary schools (4-year general, 5-year technical secondary and sectoral vocational schools) and higher education units. The schools may be public/state or non-public/none-state, which still may benefit from public funding (e.g., Salesian School Complex). They may be created by: legal bodies (associations, religious bodies, foundations, companies) and by individuals. Currently, there are approximately 500 catholic schools in Poland. It is however interesting, as despite such a high percentage of the population of Poland belonging to the Catholic Church, only a small percentage of children and adolescents attend Catholic schools. In Poland, 70,000 students attend Catholic schools, including 21,000 high school students (4.5 percent of the total number of students in this type of school). This suggests that the choice of a Catholic school is a deliberate religious and moral decision of parents and children, which may suggest a specific axiological identity of students of Catholic schools in Poland. Moreover, most Catholic schools require a so called Catholic faith declaration (as in the case of the Salesian School Complex).

The aim of this study was to examine the influence of the pandemic of COVID-19 on religious practices and to verify how religiosity moderated the influence of the pandemic on family interactions and attitudes toward health issues in adolescents.

## 2. Materials and Methods

### 2.1. Participants and Procedure

The study groups included 561 adolescent high school students (314 females and 247 males) who volunteered to take part in our study. Their ages ranged from 16 to 19 years (mean  = 16.25 years , SD  =  1.15 years). According to the aim of the study, to include religious adolescents we invited students from two kinds of high schools: public and Catholic. Directors of the schools approved the on-line questionnaire and asked students to voluntarily and anonymously participate in the study. The sociodemographic characteristics of the study samples are presented in Table 1. The study was conducted during September and October 2021, after the third wave of COVID-19. During all waves, pandemic restrictions implemented by the Polish Ministry of Health involved religious practices, as they included the cancellation of pilgrimages, closure of religious educational institutions, and restrictions on the number of participants during mass (5 persons in March 2020, then 2-m distance between attendants from July 2020, then 1 person per 15 m^2^ and at least 1.5 m distancing during the second and third wave, with the obligation to cover nose and mouth with a mask). During the third wave, social functioning was limited (hotels, galleries, cinemas, theatres, and sport facilities were closed), but ‘hard lock-down’ (implemented during the first wave) with only on-line education was not repeated [17,21,22].

Participants were recruited in four public high schools from Lodz and a nearby city (Piotrków Trybunalski) (partner high schools of Medical University of Lodz) and four Catholic (non-public) schools from east Poland: Lodz, Ostróda, Legionowo, and Mińsk Mazowiecki) (Salesian School Complex). The choice of the schools was related to certain public school partnerships with the researchers’ university, which helped to cooperate with school directors. The Salesian School Complex is the biggest network of Catholic schools and its structure involves division into four quadrants (regions) from which we asked the east-central inspectorate to take part in the study as it involved Lodz and the surrounding cities. The students were asked by teachers to voluntarily fill in an on-line questionnaire (designed with Google Forms) after classes. They were provided with necessary information and explanations (about the aim of the study, etc.) by the teachers and this information was also available when they entered the link to the survey. The students were informed that they would be anonymously asked about their and their families’ religiosity and family relations and that filling in the questionnaire would not make it possible to identify them. The questionnaire was designed in such a way that only full responses to each section of questions enabled them to proceed with the questionnaire. Those who regarded questions as too intimate could leave such a comment and opt out, but they could not fill in further questions so their partial responses were not included in the analysis. Only fully completed questionnaires could have been sent to the database. Participants who had any further comments (e.g., related to the aim of the study, the form’s questions, etc.) at the end of the form were able to write and send it to the research assistants.

The study protocol was reviewed by the Bioethical Committee of the Medical University of Lodz, and due to the voluntary, anonymous, and on-line character of the study, it was deemed that the protocol was not subject to the Committee’s approval (RNN/210/21/KE). All of the participants were informed by teachers that the data would be gathered anonymously and only for research purposes, and that withdrawal of participation was possible at any moment of the survey.

### 2.2. Questionnaire

#### 2.2.1. Demographic Data

The first set of questions concerned: age, type of high school (public, Catholic), the city of living, number of brothers and sisters, and a question about the number of people in the household with whom the respondent lived; mother, father, siblings, grandparents.

#### 2.2.2. Religiosity and Its Change during the Pandemic

To measure a level of religiosity with an objective criteria we included questions concerning participants and their family’s religious practices. There were two questions concerning this aspect: How often before the pandemic did you attend a mass in a church and/or school (everyday/a few (2–3) times a week/once a week/once a month/occasionally or I did not attend a mass, where the first three answers qualified a respondent as ‘religious’). We also asked the same question, but concerning the participant’s mother, father, grandmother, and/or grandfather with the same possible answers. Further, a question concerning change of religious practices during the pandemic was asked: How has the pandemic changed your attendance on masses (I attend: with the same frequency and the same form/with the same frequency, but changed form (internet, TV)/less frequently, but the same form/less frequently and changed form or I do not attend) (it was specified that this question was related to time outside a short period when churches were closed due to pandemic restrictions).

#### 2.2.3. The Pandemic’s Influence on Religiosity and Family Interactions

To verify the influence of the COVID-19 pandemic on these aspects, we included several items with the possible answers rated on a five-point Likert scale, ranging from 1 (very strongly disagree) to 5 (very strongly agree). The items were: ‘The pandemic caused my faith to be stronger’, ‘The pandemic caused the faith in my family to be stronger’, ‘The pandemic caused my family relations to be stronger’, ‘During the pandemic my family spent more time together, for example, playing games or talking’, ‘During the pandemic there were more arguments among members of my family’, ‘The pandemic caused me to become more concerned about the health of my family members’. The students were asked to report the changes in the mentioned aspects during the pandemic regardless of the initial state, which could have also meant lack of faith, relations, etc., before the pandemic.

#### 2.2.4. Attitude to Health Issues

To check adolescents’ attitude to health issues we asked them to rate the following items on a five-point Likert scale, ranging from 1 (very strongly disagree) to 5 (very strongly agree): ‘If I was older, I would volunteer to help patients suffering from COVID-19′, ‘I am aware of the chronic diseases my parents suffer from’, ‘I am aware of the chronic diseases my grandparents suffer from’ (regardless of their having any), ‘I am interested in the health issues concerning members of my family’, ‘I often ask my relatives about their health’, ‘Believers get sick less often’, ‘Believers can accept the disease easier than non-believers’, ‘I encourage parents/grandparents to undergo preventive screening tests’.

### 2.3. Statistical Methods

To verify the significance of the correlations between the analyzed variables, we used the chi-square test. In order to assess which responses influenced the results, we calculated the residuals, that is, standardized differences between the actual number of given responses and their theoretical number (the difference between the actual value and the predicted value from the line of best fit), for instance, what it would be if there was no relationship between the analyzed variables. Cases where the number of responses was greater/less than the theoretical number were marked in Table 2, Table 3 and Table 4 (answers given more/less often than predicted in relation to their theoretical number). For statistical purposes some answers were grouped: “attendance in the mass” as “no” (“no”, “I do not know/I do not have an opinion”), “rarely” (“once a month”, “once per 2 weeks”, “occasionally”), “once a week” and “frequently” (“everyday”, a few times per week”) and “multi-generationality” defined as living with at least one grandparent. All tests were two-tailed at a significance level of *p* < 0.05. All statistical analyses were performed using the STATISTICA PL package.

## 3. Results

Basic demographic characteristics (mean age, number of brothers and sisters, living in a multi-generation house) did not differ between students from Catholic and public schools. Students from Catholic schools significantly more often attended a mass regularly before the pandemic and also during the pandemic (Table 1).

Students from Catholic schools also answered differently to a number of questions concerning the pandemic’s influence on religiosity and family interactions and health issues. They more often answered affirmatively that the pandemic made their and their family’s faith stronger and less often than predicted they denied that believers get sick less often. Also, more often they agreed that believers can easier, than non-believers, accept the disease. On the other hand, Catholic school students encourage parents and grandparents to undergo preventive screening tests less often than public school students. The type of school did not influence family interactions during the pandemic, willingness to voluntarily help patients suffering from COVID-19, nor their attitude to family health issues during the pandemic (Table 2).

We also analyzed how answers to the questions differ between students attending a mass with different frequency. We accepted this variable as a measure of being ‘a believer’. It was found that two groups of students answered affirmatively to a question concerning the pandemic’s influence on their and their family’s faith to become stronger—those who did not attend a mass before the pandemic and those who did it regularly/once a week. Interestingly, more often than predicted, those who attended a mass more frequently than once a week negated that the pandemic caused their family relations to become stronger or that their family spends more time together. This was however true for ‘non-believer’ students. A similar observation was made in the case of the pandemic’s influence on being more concerned about the health of students’ family members. Less frequently than predicted, ‘non- believers’ negated the will to help patients suffering from COVID-19. More frequently than predicted ‘non-believers’ responded affirmatively that believers can accept the disease easier than non-believers. There were no differences in questions concerning attitude to family health issues (Table 3).

We also considered living in a multi-generational house as a possible factor influencing attitudes towards the pandemic and health issues. It, however, influenced responses to only two similar questions. It appeared that students who do not live in a multi-generational house more frequently negated that believers get sick less often and that believers can accept the disease easier than non-believers contrary to those living in such a house (Table 4).

## 4. Discussion

The aim of the present study was to examine how the pandemic of COVID-19 changed religious practices and how religiosity moderated the influence of the pandemic on family interactions and attitudes towards health issues in families of adolescents. Our findings showed that most Catholic school students have not changed their religious practices during the pandemic or just changed the form of attendance to TV or internet, while most public school students had not attended a mass before and during the pandemic. However, in both groups of students there was a percentage of those who stopped religious practices during the pandemic (8.6% of catholic school students versus 12.9% from public schools). This may also be discussed in the aspect of the report presented by Gawroński et al. 2021. Based on their fair literature analysis and interpretation, the authors found that in recent years persistent critical opinions towards the Catholic Church appeared. This suggests that the Church in Poland has been losing confidence in its image, probably due to inefficient communication [23]. Attending Catholic school and being a practicing believer influenced some aspects of faith and family relations during the pandemic, but in most aspects, they did not influence attitudes toward health issues. More often than predicted, students who regularly (once a week) attended a mass were willing to volunteer to help patients suffering from COVID-19, while those who attended more often than once a week more often denied a willingness to volunteer during the pandemic. To our knowledge, this is the first study examining interactions between religiosity and the influence of COVID-19 on family issues related to health among Polish adolescents during the pandemic.

Although there are studies concerning the influence of the pandemic of COVID-19 on family and social issues, in this study, we checked how it was mediated by religiosity. We found that most adolescents negated that the pandemic caused their family relations to be stronger (33% affirmative answers), however affirmative residuals were observed for definitely affirmative responses in the case of the ‘non-believer’ (not attending a mass at all). Moreover, some students reported that they do spend more time with their families on activities (30% affirmative answers), but affirmative answers were more frequent in ‘non-believers’. This may result from the fact that adolescents experienced some distress during the pandemic and isolation from peers and friends. In this situation, they were looking for some support—‘believers’ could have found it in faith, and ‘non-believers’ in family members (with whom they were ‘lock-downed’). As shown in many previous studies religious beliefs can help to deal with psychological stress and trauma (during COVID-19, other pandemics and/or disasters) [24,25,26,27].

Most students appeared to be interested in their parents’ health and the pandemic did not make them more concerned about the health of family members (38% negative and 41% affirmative answers). Interestingly, the same number of participants answered affirmatively and negatively to a question concerning encouraging parents/ grandparents to undergo preventive screening tests, but definitely affirmative answers were more frequent for public school students. This indicates that in both types of schools more educational programs concerning health prophylaxis is needed. However, ‘non-believers’ significantly more frequently answered affirmatively and ‘strong believers’ (frequently practicing)—less frequently gave the same answer. This may result from their belief in other ‘non-physical’ determinants of health. On the other hand, this group denied that their faith became stronger during the pandemic, contrary to students who did not attend a mass (regarded as ‘non-believers’) and those attending once a week—most of them claimed affirmatively to this statement. This was also a common statement in Catholic school students, which could have resulted from direct support from Catholic school teachers on how to deal with the distress caused by the pandemic. This observation could be discussed in light of the results presented by Kowalczyk et al., (2020), who focused on the adult Polish population and concluded that people who face fear or illness often experience a ‘spiritual renewal’ [16]. However, the possibility of applying these results to the adolescent population needs verification in further studies. Moreover, there is a report linking religious attendance with loneliness and mental well-being via social network size. The authors found that the association between religion and social functioning is not only related to interactions within the religious community, and that the impact of religiosity on social ties can be more nuanced [28].

Regarding students’ attitudes towards the pandemic, only 27% answered that if they were older, they would volunteer to help patients suffering from COVID-19, and answers were mediated by religiosity—those attending a mass once a week were willing to volunteer while those attending more frequently more often answered negatively. This may result from the fact that ‘regular’ believers guided their answers with altruistic Catholic beliefs, while those ‘strong believers’ may more rely on prayer than physical altruistic actions. It would be worthwhile to further examine the differences in attitude to different health and social aspects in relation to the “strength” of faith and search for the possible explanations of these differences. The study by Domaradzki and Walkowiak (2021) showed that medical students’ religiosity was not a significant factor for deciding to volunteer during the pandemic of COVID-19, but it guided their motivation. The students had altruistic and self-interested motivations, but altruistic motivations were more common among religious respondents [29]. Additionally, the educational system proposed by Galiatsatos et al. (2021), Just-In-Time COVID-19 Training in Catholic Schools, could increase population health strategies being understood and accepted and increase the rate of adolescents willing to cooperate and be ready to volunteer [30].

Attitudes toward the pandemic mediated by religiosity may result from the beliefs of believers getting sick less often and being able to more easily accept the disease than non-believers. Although these statements were denied by most students, they were more often denied by frequently attending believers, but accepted by those who attend a mass once a week. Catholic school students more rarely denied these statements. Apart from the influence of religiosity, it appears that students who lived in multi-generational houses less frequently denied these statements. This may result from the influence of grandparents who are mostly Catholic (a high percentage of believers in this generation in Poland) and being constantly present during up-bringing could have inspired such belief in children. This may also explain the different influences of school and religiosity showing familial background to be most important.

### Study Assessment

There are some limitations of the study that may impact generalizability and interpretation of the results. First, students from only central and north-east Poland were included in the study. Also, as many students did not complete the questionnaire, the results cannot be generalized for not even the entire Polish adolescent population of the aforementioned regions. The main strength of the study is that we reported the influence of adolescents’ religiosity on different aspects of life (related to family and health issues) during a COVID-19 pandemic in Poland, which was not examined earlier. This report may stimulate further studies on the topic.

## 5. Conclusions

To sum up, to our knowledge this is the first report presenting the influence of religiosity and attending different schools on attitudes towards the pandemic, family relations, and different health aspects during the pandemic. We showed that in some adolescents’ opinions the pandemic caused family relations to be stronger, however this effect was modified by religiosity as more frequently attending believers less often reported such an influence. It may be due to the fact that religious families, usually favoring the family, could have had better integration before the pandemic, which results from the priority of the family in the religious value system. This highlights the need to secure, especially for non-believing adolescents, family support during the pandemic, while in believers faith may provide such support. It appears that frequently attending students differ in many aspects from those regularly (once a week) attending, which shows that not being a believer itself, but the strength of faith mediates attitudes to different life aspects. This observation should be verified in further studies, also in the aspect of health-related issues, e.g., knowledge and attitude towards prophylactic screening tests and prophylactic measures such as obligatory and voluntary vaccinations. This may help to design and properly target appropriate educational programs.

## Figures and Tables

**Table 1 ijerph-19-06462-t001:** Detailed information on the participants taking into consideration the type of school (Catholic versus public).

Characteristic	Type of School		
	Catholic *n* = 266	Public *n* = 295	Chi/*p*
Sex			
Male	122 (46%)	125 (42%)	ns
Female	144 (54%)	170 (58%)	
Mean age [years]	16.22. SD 1.3	16.22. SD 1.3	ns
Place of living			
City 50–500 k citizens	94 (35%)	57 (19%)	18.24
City > 500 k citizens	172 (65%)	238 (81%)	<0.0001
Number of sisters/brothers [mean]	0.65/0.76	0.59/0.75	ns
Living in multi-generational house	20 (7.5%)	30 (10.2%)	ns
Attending a mass before the pandemic			
No	27 (10.2%)	131 (44.4%)	116.94 <0.0001
Rarely	57 (21.4%)	83 (28.1%)	
Once a week	151 (56.8%)	77 (26.1%)	
Frequently	31 (11.6%)	4 (1.4%)	
Attending a mass during the pandemic			
Changed to TV/internet attendance	69 (25.9%)	29 (9.8%)	90.12 <0.0001
Less frequently than before the pandemic	57 (21.4%)	40 (13.6%)	
No differences in frequency or more often	90 (33.8%)	57 (19.3%)	
Not attending during the pandemic	50 (18.8%)	169 (57.3%)	

**Table 2 ijerph-19-06462-t002:** Comparison of responses to questions about religion, the pandemic, and health issues between Catholic and public high school students.

	School		Chi/*p*
	Catholic *n* = 266	Public *n* = 295	
The pandemic caused my faith to be stronger (381 vs. 105 vs. 75) *.			
1	102 (38.3%)	209 (70.8%)	61.57 <0.0001
2	44 (16.5%)	26 (8.8%)	
3	74 (27.8%)	31 (10.5%)	
4	32 (12%)	20 (6.8%)	
5	14 (5.3%)	9 (3.1%)	
The pandemic caused the faith in my family to be stronger (380 vs. 126 vs. 55).			
1	85 (32%)	196 (66.4%)	68.72 <0.0001
2	59 (22.2%)	40 (13.6%)	
3	86 (32.3%)	40 (13.6%)	
4	23 (8.6%)	14 (4.7%)	
5	13 (4.9%)	5 (1.7%)	
I am aware of the chronic diseases my parents suffer from (82 vs. 57 vs. 422).			
1	24 (9%)	21 (7.1%)	18.457 0.001
2	13 (4.9%)	24 (8.1%)	
3	41 (15.4%)	16 (5.4%)	
4	76 (28.6%)	87 (29.5%)	
5	112 (42.1%)	147 (49.8%)	
Believers get sick less often (501 vs. 41 vs. 19).			
1	187 (70.3%)	259 (87.8%)	30.23 <0.0001
2	34 (12.8%)	21 (7.1%)	
3	33 (12.4%)	8 (2.7%)	
4	7 (2.6%)	3 (1%)	
5	5 (1.9%)	4 (1.4%)	
Believers can accept the disease easier than non-believers (311 vs. 121 vs. 129).			
1	86 (32.3%)	155 (52.5%)	34.25 <0.0001
2	35 (13.2%)	35 (11.9%)	
3	58 (21.8%)	63 (21.4%)	
4	50 (18.8%)	24 (8.1%)	
5	37 (13.9%)	18 (6.1%)	
I encourage parents/grandparents to undergo preventive screening tests (221 vs.120 vs. 220).			
1	73 (27.4%)	54 (18.3%)	10.79 0.029
2	48 (18%)	46 (15.6%)	
3	55 (20.7%)	65 (22%)	
4	43 (16.2%)	52 (17.6%)	
5	47 (17.7%)	78 (26.4%)	
The pandemic caused my family relations to be stronger (245 vs.133 vs.183) *.			8.77/0.067
During the pandemic my family spends more time together (268 vs.125 vs.168).			7.66/0.105
During the pandemic there are more arguments among members of my family (236 vs. 121 vs. 204).			3.44/0.487
The pandemic caused me to become more concerned about the health of my family members (211 vs. 122 vs. 228).			0.62/0.961
If I was older, I would volunteer to help patients suffering from COVID-19 (290 vs. 121 vs. 150).			0.89/0.926
I am aware of the chronic diseases my grandparents suffer from (130 vs. 102 vs. 329).			3.75/0.441
I am interested in the health issues concerning members of my family (47 vs. 63 vs. 451).			5.75/0.218
I often ask my relatives about their health (145 vs. 107 vs. 309).			6.18/0.186

Note. Full data only for significant results. Green—Affirmative residuals, blue—Negative, * answers 1–2 (“no”) vs. 3 vs. 4–5 (“yes”) in the whole group.

**Table 3 ijerph-19-06462-t003:** Comparison of responses to questions about religion, the pandemic, and health issues between high school students depending on their religious practices.

	Attending a Mass in the Church before the Pandemic				Chi/p
	No	Rarely	Once a Week	Frequently	
The pandemic caused my faith to be stronger.					
1	74 (32.5%)	82 (58.6%)	4 (11.4%)	151 (95.6%)	
2	38 (16.7%)	21 (15%)	7 (20%)	4 (2.5%)	
3	64 (28.1%)	29 (20.7%)	11 (31.4%)	1 (0.6%)	195.25
4	38 (16.7%)	6 (4.3%)	8 (22.9%)	0 (0%)	<0.0001
5	14 (6.1%)	2 (1.4%)	5 (14.3%)	2 (1.3%)	
The pandemic caused the faith in my family to be stronger.					
1	67 (29.4%)	72 (51.4%)	7 (20%)	135 (85.4%)	
2	51 (22.4%)	34 (24.3%)	6 (17.1%)	8 (5.1%)	
3	75 (32.9%)	28 (20%)	12 (34.3%)	11 (7%)	168.93
4	26 (11.4%)	5 (3.6%)	3 (8.6%)	3 (1.9%)	<0.0001
5	9 (3.9%)	1 (0.7%)	7 (20%)	1 (0.6%)	
The pandemic caused my family relations to be stronger.					
1	55 (24.1%)	33 (23.6%)	7 (20%)	61 (38.6%)	
2	37 (16.2%)	27 (19.3%)	3 (8.6%)	22 (13.9%)	
3	57 (25%)	32 (22.9%)	6 (17.1%)	38 (24.1%)	28.39
4	49 (21.5%)	37 (26.4%)	12 (34.3%)	30 (19%)	0.005
5	30 (13.2%)	11 (7.9%)	7 (20%)	7 (4.4%)	
During the pandemic my family spends more time together.					
1	61 (26.8%)	37 (26.4%)	5 (14.3%)	65 (41.1%)	
2	34 (14.9%)	28 (20%)	8 (22.9%)	30 (19%)	
3	52 (22.8%)	37 (26.4%)	6 (17.1%)	30 (19%)	29.96
4	46 (20.2%)	29 (20.7%)	11 (31.4%)	22 (13.9%)	0.003
5	35 (15.4%)	9 (6.4%)	5 (14.3%)	11 (7%)	
The pandemic caused me to become more concerned about the health of my family members.					
1	39 (17.1%)	17 (12.1%)	9 (25.7%)	51 (32.3%)	
2	38 (16.7%)	24 (17.1%)	3 (8.6%)	30 (19%)	
3	43 (18.9%)	44 (31.4%)	7 (20%)	28 (17.7%)	36.56
4	67 (29.4%)	34 (24.3%)	13 (37.1%)	34 (21.5%)	0.007
5	41 (18%)	21 (15%)	3 (8.6%)	15 (9.5%)	
If I was older, I would volunteer to help patients suffering from COVID-19.					
1	59 (25.9%)	41 (29.3%)	9 (25.7%)	71 (44.9%)	
2	45 (19.7%)	33 (23.6%)	6 (17.1%)	26 (16.5%)	
3	52 (22.8%)	28 (20%)	6 (17.1%)	35 (22.2%)	27.27
4	39 (17.1%)	20 (14.3%)	5 (14.3%)	13 (8.2%)	0.007
5	33 (14.5%)	18 (12.9%)	9 (25.7%)	13 (8.2%)	
Believers get sick less often.					
1	164 (71.9%)	118 (84.3%)	14 (40%)	150 (94.9%)	
2	34 (14.9%)	11 (7.9%)	7 (20%)	3 (1.9%)	
3	22 (9.6%)	9 (6.4%)	7 (20%)	3 (1.9%)	92.84
4	5 (2.2%)	2 (1.4%)	2 (5.7%)	1 (0.6%)	<0.0001
5	3 (1.3%)	0 (0%)	5 (14.3%)	1 (0.6%)	
Believers can accept the disease easier than non-believers.					
1	59 (25.9%)	64 (45.7%)	4 (11.4%)	114 (72.2%)	
2	26 (11.4%)	21 (15%)	2 (5.7%)	21 (13.3%)	
3	63 (27.6%)	33 (23.6%)	6 (17.1%)	19 (12%)	148.51
4	47 (20.6%)	13 (9.3%)	10 (28.6%)	4 (2.5%)	<0.0001
5	33 (14.5%)	9 (6.4%)	13 (37.1%)	0 (0%)	
During the pandemic there were more arguments among members of my family.					17.88/0.12
I am aware of the chronic diseases my parents suffer from.					16.96/0.151
I am aware of the chronic diseases my grandparents suffer from.					10.56/0.567
I am interested in the health issues concerning members of my family.					18.99/0.089
I often ask my relatives about their health.					12.24/0.427
I encourage parents/grandparents to undergo preventive screening tests.					5.28/0.948

Note. Full data only for significant results. Green—Affirmative residuals, blue—Negative.

**Table 4 ijerph-19-06462-t004:** Comparison of responses to questions about religion, the pandemic, and health issues in relation to living in a multi-generational house.

	Living in a Multi-Generational House		Chi/*p*
	No *n* = 483	Yes *n* = 78	
Believers get sick less often.			
1	392 (81.2%)	54 (69.2%)	
2	47 (9.7%)	8 (10.3%)	
3	30 (6.2%)	11 (14.1%)	9.63
4	7 (1.4%)	3 (3.8%)	0.047
5	7 (1.4%)	2 (2.6%)	
Believers can accept the disease easier than non-believers.			
1	210 (43.5%)	31 (39.7%)	
2	66 (13.7%)	4 (5.1%)	
3	105 (21.7%)	16 (20.5%)	11.15
4	61 (12.6%)	13 (16.7%)	0.025
5	41 (8.5%)	14 (17.9%)	
All other questions concerning religion, pandemics, and health issues—ns			

Note. Full data only for significant results. Green—Affirmative residuals, blue—Negative, ns—Not significant.

## Data Availability

Data available on request from the corresponding author.

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
