# Peer review of "The Influence of Faith and Religion on Family Interactions and Interest in Health Issues during the COVID-19 Pandemic—A Study among Polish Adolescents"

_ijerph, 2022, doi:10.3390/ijerph19116462_

Round 1
Reviewer 1 Report
The paper under consideration is about whether and how, during the third wave of Covid-19, faith and religion influenced Polish adolescents’ religious practices, family relations, and attitudes towards different health aspects (190). The research does not deal with other effects of the pandemic in young people, such as mental health, or behavioral changes, for which numerous studies are available (204). Materials, methods and results of the study, discussion on the findings, and conclusions about the research, are offered in the paper.
If the authors had known that some researches have been devoted to analyze pandemic changes on Polish adolescents’ religiosity, they affirm that “no previous studies have explored the potential effect of faith and religion” on family interactions and attitudes to health issues (45). In studying this aspect lies the novelty of the article (201). However, the authors also recognized that, even if “scarce”, there is literature concerning religiosity’s influence on “family dynamics during the pandemic” (214). As a matter of clarification, are adolescents included in the researches sustaining this literature? In this case, could it be said that some studies, even scarce, have explored the potential effect of religiosity on family interactions?
As far as I can see, some information may be clarified. For example,
Line 72-73: “They were provided with necessary information and explanations”. Who are “they”? Students or teachers? In fact, both were mentioned in the previous lines.
Line 215-218: “Poland is regarded as a Catholic country of about 94% declaring themselves as ‘believers’ and about 80% of society declaring themselves as ‘regularly or irregularly practicing believers’, but with about 40% of those in the 16-24 years age group”. A 40% declaring themselves believers? Practicing believers? To what age can be referred the previous 94% and 80%?
Line 220-222: “The previous study concerning this aspect showed that meaning-making and fear of COVID-19 influenced the relationship between religiosity and well-being among late adolescents”. What exactly is the previous study?
In line 297 it is said: since “many students did not complete the questionnaire, the results cannot be generalized for the entire Polish adolescent population”. Maybe it should be better said, “for not even the entire Polish adolescent population of the aforementioned regions”. Clearly, one cannot nationalize the results of a regional sample.
The study of Kowalczyk et al (2020) about “spiritual renewal” is mentioned twice. In line 52 it is said that people “experiencing fear and illness may be subjected to the ‘spiritual renewal’”; in line 264, it is said however that “people experiencing fear, suffering, or illness often experience a ‘spiritual renewal’”. Therefore, people may experience or often experience? Besides, in which sense a study focused on “the adult Polish population” (263) can be used to interpret what it is happening with adolescents (262)?
Regarding elements of style, perhaps the subtitles on lines 91, 109 and 118 can be highlighted adjusting the spacing before. Moreover, some titles on Table 1 — “Attending a mass before the pandemic” and “Attending a mass during the pandemic” — as well as on Table 2 — “Believers get sick less often” — have to be centered. In lines 295-301, the authors mention limitations but also a strength and a prospect. For this reason, could the title be “Study assessment” instead of “Study limitations”?
Author Response
Dear Reviewer,
Thank you for your interest in our manuscript entitled " The influence of faith and religion on family interactions and interest in health issues during the COVID-19 pandemic - a study among Polish adolescents”. We would like to thank anonymous Reviewers for their detailed valuable comments and reviews, which helped to improve the manuscript. In this revision we addressed all your comments (in the text changes are marked in red colour). We hope that this revision meets with your approval.
Sincerely,
The Authors
Responses to Reviewer’s comments:
Reviewer 1
1.If the authors had known that some researches have been devoted to analyze pandemic changes on Polish adolescents’ religiosity, they affirm that “no previous studies have explored the potential effect of faith and religion” on family interactions and attitudes to health issues (45). In studying this aspect lies the novelty of the article (201). However, the authors also recognized that, even if “scarce”, there is literature concerning religiosity’s influence on “family dynamics during the pandemic” (214). As a matter of clarification, are adolescents included in the researches sustaining this literature? In this case, could it be said that some studies, even scarce, have explored the potential effect of religiosity on family interactions?
Thank you for this comment – it has been clarified what is “new”, “scarce” and what is the novelty of the paper. As you mentioned there have been studies concerning pandemic influence on psychical health and general well-being of polish adolescents and studies regarding religiosity influence on adolescents’ well-being during pandemic.
“Although some studies have explored the potential effect of faith and religion on the positive and/or negative influence of the pandemic on family interactions, there are no researches concerning their effect on attitudes to family health issues among Polish adolescents during the COVID-19 pandemic.”
2.Line 72-73: “They were provided with necessary information and explanations”. Who are “they”? Students or teachers? In fact, both were mentioned in the previous lines.
This was explained:
“Students were provided with necessary information and explanations (about the aim of the study etc.) by the teachers and these information was also available when they entered the link to the survey.”
3.Line 215-218: “Poland is regarded as a Catholic country of about 94% declaring themselves as ‘believers’ and about 80% of society declaring themselves as ‘regularly or irregularly practicing believers’, but with about 40% of those in the 16-24 years age group”. A 40% declaring themselves believers? Practicing believers? To what age can be referred the previous 94% and 80%?
The percentages and their meaning were clarified:
“Poland is regarded as a Catholic country of about 87% (2021) - 94% (1992) declaring themselves as ‘believers’ and about 76% of society declaring themselves as “regularly or irregularly practicing believers” (2021) (regardless age, over 18 years old), but with about 36% of “non-practicing” in the 18-24 years age group (data from the Centre for Public Opinion Research, 2021).”
4.Line 220-222: “The previous study concerning this aspect showed that meaning-making and fear of COVID-19 influenced the relationship between religiosity and well-being among late adolescents”. What exactly is the previous study?
I am not sure if I understand your point. “The previous” was changed to specify the study:
“The study by Krok et al. (2021) concerning this aspect…”
5.In line 297 it is said: since “many students did not complete the questionnaire, the results cannot be generalized for the entire Polish adolescent population”. Maybe it should be better said, “for not even the entire Polish adolescent population of the aforementioned regions”. Clearly, one cannot nationalize the results of a regional sample.
This was specified as you suggested:
“First, students from only central and north-east Poland were included in the study. Also, as many students did not complete the questionnaire, the results cannot be generalized for not even the entire Polish adolescent population of the aforementioned regions.”
6.The study of Kowalczyk et al (2020) about “spiritual renewal” is mentioned twice. In line 52 it is said that people “experiencing fear and illness may be subjected to the ‘spiritual renewal’”; in line 264, it is said however that “people experiencing fear, suffering, or illness often experience a ‘spiritual renewal’”. Therefore, people may experience or often experience? Besides, in which sense a study focused on “the adult Polish population” (263) can be used to interpret what it is happening with adolescents (262)?
The authors originally stated that “Our research conducted as part of the survey indicates that people experiencing fear, suffering or illness often experience a “spiritual renewal.” so “often” was kept as “original”. We mentioned that the study involved adults, although authors’ conclusion involves generally “people”. This mechanism could also apply to adolescents, which however should be verified in future studies – this was clarified in the text:
“This observation could be discussed in light of the results presented by Kowalczyk et al, (2020) who focused on the adult Polish population and concluded that “people experiencing fear, suffering, or illness often experience a ‘spiritual renewal’ [14]. However, the possibility of applying these results to adolescent population needs verification in further studies.”
7.Regarding elements of style, perhaps the subtitles on lines 91, 109 and 118 can be highlighted adjusting the spacing before. Moreover, some titles on Table 1 — “Attending a mass before the pandemic” and “Attending a mass during the pandemic” — as well as on Table 2 — “Believers get sick less often” — have to be centered. In lines 295-301, the authors mention limitations but also a strength and a prospect. For this reason, could the title be “Study assessment” instead of “Study limitations”?
Done as suggested
Reviewer 2 Report
The paper deals with relevant issues to public health, which are particular important at the pandemic times we continue to live. However, it has both serious methodological flaws and several biased and hardly neutral assumptions that undermine the scientificity and quality of the study and support my recommendation of rejection.
1. The study objectives are unclear and some questions suffer from a problem of logical causality. Do the authors want to understand the consequences of Covid 19 on family relationships and on adolescents' attitudes toward health issues? Or the role of religion as a mediating variable in assessing these consequences? Or whether adolescents have changed their religious practices? The formulation is confused and it is not clear exactly what the authors intend to explain.
2. In the section on methods and materials, there are serious flaws in the items included in the questionnaire, which compromise the results observed.
- Given the obvious importance of socio-economic status in the consequences of Covid 19 (which, by the way, the authors mention in the introduction), it is difficult to understand why, in the demographic data, no question was included to characterize the families of adolescents in terms of their socio-economic status.
- Regarding the questions on religiosity and its change during the pandemic, there are items that are not valid. For example 'The pandemic caused my faith to be stronger' and 'The pandemic caused the faith in my family to be stronger'. These statements make the wrong assumption that there was faith prior to the pandemic. How does an adolescent whose faith before the pandemic was absent would answer this question?
- Regarding the questions on attitude to health issues, there are also items that are not valid. ‘I am aware of the chronic diseases my parents suffer from’ and ‘I am aware of the chronic’ are assertions that assume that parents and other household members have chronic diseases. How do adolescents whose relatives do not have chronic diseases can answer?
3. The article is full of moral judgments about Catholicism, as if there was an intrinsic superiority of religion as a way of mitigating the consequences of Covid 19, which is not acceptable in a scientific article. Moreover, it draws simplistic conclusions that are not supported by empirical data.
Here are some examples:
“In this situation they were looking for some support – ‘believers’ could have found it in faith, and ‘non-believers’ in family members (with whom they were ‘lock-downed’).” (lines 242-244). This statement has no evidence to support it. It is a simplistic statement that presents a dichotomized view of support as either one thing or the other. Why can't believers also have had family support? Does the fact that they have faith support imply that family is excluded? And can only family and faith support be provided? What about other people in the sociability networks, e.g. friends? These questions are completely absent from this study.
“This was also a common statement in Catholic school students, which could have resulted from a direct support from Catholic school teachers on how to deal with the distress caused by the pandemic” (lines 259-261). Why is it assumed that in the Catholic school there was support and in the public school there was not? The authors do not present data to demonstrate this.
“This may result from the fact that ‘regular’ believers guided their answers with altruistic Catholic beliefs while those ‘strong believers’ may more rely on prayer than physical altruistic actions.” (lines 272-275). This explanatory hypothesis is not justified.
“It may be due to the fact that Catholic families had better integration before the pandemic, which results from the priority of the family in the religious value system” (lines 307-309). Again, a moral claim about the Catholic religion, as if it was the key factor that guarantees a better integration of families.
Author Response
Dear Reviewer,
Thank you for your review of our manuscript entitled " The influence of faith and religion on family interactions and interest in health issues during the COVID-19 pandemic - a study among Polish adolescents”. Although it is difficult (or even impossible) to answer all your criticism, thank you for your time and effort to prepare this review. According to the Editor’s suggestion we answered the following items:
- Regarding the questions on religiosity and its change during the pandemic, there are items that are not valid. For example 'The pandemic caused my faith to be stronger' and 'The pandemic caused the faith in my family to be stronger'. These statements make the wrong assumption that there was faith prior to the pandemic. How does an adolescent whose faith before the pandemic was absent would answer this question?
This question aimed to examine “the change” in faith caused by pandemic, not religiosity itself (there were questions concerning attendance in a mass before and during pandemic with the possible answer: I do/did not attend). So, if someone was not “a believer” and still is not a believer would answer “definitely not”, if he/she was a “strong believer” and during pandemic is still “the same strong” believer he/she would also chose “1 – definitely not” as in both “the change” is similar (none). Similarly, in case of “The pandemic caused my family relations to be stronger” – these relations did not have to be regarded as strong before pandemic to answer the question (like in most of thr questions in this section). Before releasing the questionnaire in all schools included in the study we conducted a pilot study in one of the classes and at the end of the survey the respondents were asked (also anonymously) for any comments concerning problems, comments, suggestions regarding the questionnaire. We kept this “commentary section” during the whole study and none of the students reported problems with answering these questions.
This was probably not clearly explained in the ms, so we clarified this:
“The students were asked to report the changes in the mentioned aspects during pandemic regardless the initial state, which could have also meant lack of faith, relations, etc. before pandemic.”
- Regarding the questions on attitude to health issues, there are also items that are not valid. ‘I am aware of the chronic diseases my parents suffer from’ and ‘I am aware of the chronic’ are assertions that assume that parents and other household members have chronic diseases. How do adolescents whose relatives do not have chronic diseases can answer?
I think the answers above also apply to this comment. To answer your question: they would answer that they were aware about the chronic diseases in their family (that their parents did not have any chronic diseases). Maybe this subtle difference is not obvious in our description of the questions. What we wanted to get to know was if they knew about the chronic diseases in their family (regardless their presence), not to evaluate the true health status of their family members.
To further clarify these issues, according to the Editor’s suggestion we discussed this problem in the limitations:
“Additionally, there could be some concerns about validity of some questions included in the questionnaire, especially those concerning changes in students’ and their families’ religiosity. However it should be highlighted that the questions applied to the change, so the students could answer the questions regardless their initial status (e.g. lack of faith, being non-religious). Similarly when asked about their knowledge about chronic diseases in their relatives, they could answer the question regardless the true presence of such diseases in their family as the idea was to test their awareness.” – to further comment on this, there is a difference between “I do not know anything about chronic diseases in my family” and “I know that there are no chronic diseases”,
Reviewer 3 Report
The article is interesting but needs some significant improvements.
1.In the reviewer's opinion it is not professional to formulate the title of the article in the form of a question.
2.The aim of the article expressed in lines 55 and 56 should be corrected either. In the reviewer's opinion it should not be formulated in such a way that it contains the word "whether".
3.The author writes that the study was conducted after the third wave of the pandemic. The question arises whether the period given refers to the pandemic or to the research.
4.In the reviewer's opinion, it is necessary to present what the restrictions were during the third wave of the pandemic, especially those concerning religious practices. The text can be placed as soon as information about the research is given. A lot of details are given in state documents (The Journal of Laws of the Republic of Poland. See: https://isap.sejm.gov.pl). There are also quite a few publications on this topic of the pandemic itself, religious practices and the restrictions themselves. See: RafaÅ‚ Boguszewski, Marta Makowska, Marta Bożewicz, Marta Makowska, „The COVID-19 Pandemic’s Impact on Religiosity in Poland”, Religions 11, no 12 ( December 2020): 646; Grzegorz Ignatowski, The Ambivalent Attitude of the Catholic Church in Poland Towards the COVID-19 Pandemic, Euxeinos, Vol. 12, Issue 33: 46-56; Barbara Przywara, Andrzej Adamski, Andrzej KiciÅ„ski, Marcin Szewczyk, Anna Jupowicz-Ginalska, Online Live-Stream Broadcasting of the Holy Mass during the COVID-19 Pandemic in Poland as an Example of the Mediatisation of Religion: Empirical Studies in the Field of Mass Media Studies and Pastoral Theology, Religions 12 (April 2021): 261.
5.In line 84 we read about the Ethics Committee of Medical University of Lodz and in line 320 about the Bioethical Committee. These discrepancies should be clarified as they relate to the same document.
6.In Table 1 we find data on attendance at mass in relation to Catholic and public schools. The reviewer asks for clarification on how the data on Catholic schools should be interpreted: Attending a mass before the pandemic (No 10.2%) and attending a mass during the pandemic (Not attending during the pandemic 18.8%). The question should be asked how these data relate to the information contained in the abstract and lines 193-195.
It is worth comparing the results of the study and the information on Catholics with studies conducted by the Institute for Catholic Church Statistics. See also: Gawroński, Sławomir, Dariusz Tworzydło, and Bajorek Kinga. 2021. Determinants for the Development of the Activity of the Catholic Church in Poland in the Field of Social Communication. Religions 12: 845.
7.In lines 228-229 the authors write about Muslims and refer to a specific article on Turkey. Please justify how far this is justified.
8.Please explain in the research methodology what guided the selection of specific schools and to what extent the selection of the research group was purposeful.
9.It is necessary to write on what basis the article draws the conclusions in lines 307 and 308.
10. In the conclusion it is worth indicating what are the prospects for further research.
Author Response
Dear Reviewer,
Thank you for your interest in our manuscript entitled " The influence of faith and religion on family interactions and interest in health issues during the COVID-19 pandemic - a study among Polish adolescents”. We would like to thank anonymous Reviewers for their detailed valuable comments and reviews, which helped to improve the manuscript. In this revision we addressed all your comments (in the text changes are marked in red colour). We hope that this revision meets with your approval.
Sincerely,
The Authors
Responses to Reviewer’s comments:
Reviewer 3
1.In the reviewer's opinion it is not professional to formulate the title of the article in the form of a question.
According to your suggestion we rearranged the title as a statement.
“The influence of faith and religion on family interactions and interest in health issues during the COVID-19 pandemic - a study among Polish adolescents”
2.The aim of the article expressed in lines 55 and 56 should be corrected either. In the reviewer's opinion it should not be formulated in such a way that it contains the word "whether".
Done:
“The aim of this study was to examine the influence of the pandemic of COVID-19 on the religious practices and verify how religiosity moderated the influence of the pan-demic on family interactions and attitudes towards health issues in adolescents.”
3.The author writes that the study was conducted after the third wave of the pandemic. The question arises whether the period given refers to the pandemic or to the research.
It was clarified:
“The study was conducted during September and October 2021, so after the third wave of Covid-19.”
4.In the reviewer's opinion, it is necessary to present what the restrictions were during the third wave of the pandemic, especially those concerning religious practices. The text can be placed as soon as information about the research is given. A lot of details are given in state documents (The Journal of Laws of the Republic of Poland. See: https://isap.sejm.gov.pl). There are also quite a few publications on this topic of the pandemic itself, religious practices and the restrictions themselves. See: RafaÅ‚ Boguszewski, Marta Makowska, Marta Bożewicz, Marta Makowska, „The COVID-19 Pandemic’s Impact on Religiosity in Poland”, Religions 11, no 12 ( December 2020): 646; Grzegorz Ignatowski, The Ambivalent Attitude of the Catholic Church in Poland Towards the COVID-19 Pandemic, Euxeinos, Vol. 12, Issue 33: 46-56; Barbara Przywara, Andrzej Adamski, Andrzej KiciÅ„ski, Marcin Szewczyk, Anna Jupowicz-Ginalska, Online Live-Stream Broadcasting of the Holy Mass during the COVID-19 Pandemic in Poland as an Example of the Mediatisation of Religion: Empirical Studies in the Field of Mass Media Studies and Pastoral Theology, Religions 12 (April 2021): 261.
As suggested a background concerning restrictions, mainly regarding religious practices, was added:
“The study was conducted during September and October 2021, so after the third wave of Covid-19. During all waves pandemic restrictions implemented by Polish Ministry of Health involved religious practices, as they included the cancellation of pilgrimages, closure of religious educational institutions, and restrictions on the number of partici-pants during mass (5 persons in March 2020, then 2-meter distance between attend-ants from July 2020, then 1 person per 15 m2 and at least 1.5 meter distancing during the second and third wave, with the obligation to cover nose and mouth with a mask). During the third wave social functioning was limited (hotels, galleries, cinemas, thea-tres and sport facilities were closed), but ‘hard lock-down’ (implemented during the first wave) with only on-line education was not repeated then.”
Additional references were included:
Boguszewski, R.; Makowska, M.; Bożewicz, M.; PodkowiÅ„ska, M. The COVID-19 Pandemic’s Impact on Religiosity in Poland. Religions 2020, 11, 646.
Ignatowski, G: The Ambivalent Attitude of the Catholic Church in Poland Towards the COVID-19 Pandemic, Euxeinos, 2022, 12, 46-56
Przywara, B.; Adamski, A.; Kiciński, A.; Szewczyk, M.; Jupowicz-Ginalska, A. Online Live-Stream Broadcasting of the Holy Mass during the COVID-19 Pandemic in Poland as an Example of the Mediatisation of Religion: Empirical Studies in the Field of Mass Media Studies and Pastoral Theology. Religions 2021, 12, 261.
5.In line 84 we read about the Ethics Committee of Medical University of Lodz and in line 320 about the Bioethical Committee. These discrepancies should be clarified as they relate to the same document.
This was unified as the Bioethical Committee of the Medical University of Lodz
6.In Table 1 we find data on attendance at mass in relation to Catholic and public schools. The reviewer asks for clarification on how the data on Catholic schools should be interpreted: Attending a mass before the pandemic (No 10.2%) and attending a mass during the pandemic (Not attending during the pandemic 18.8%). The question should be asked how these data relate to the information contained in the abstract and lines 193-195.
I am not sure if I have understood your point…In the abstract we stated: “Most Catholic school students have not changed their religious practices during the pandemic or just changed the form of attendance to TV or internet, while most public school students have not attended a mass during the pandemic.” According to the table 1 catholic students who have not changed their religious practices during the pandemic (33.8%) or just changed the form of attendance to TV or internet (25.9%) (which gives 60%) are the majority. 8.6% of them stopped attending a mass during pandemic. Most of public school students have not attended a mass during the pandemic (57%) and 12.9% stopped attending a mass. I guess your idea was to highlight these 8.6 vs.12.9% who stopped attending a mass. This was included in the abstract to better sum up table 1.
“Most Catholic school students have not changed their religious practices during the pandemic or just changed the form of attendance to TV or internet (59.7%). Moreover, 8.6% of them stopped the practices, in comparison with 12.9% of public school students, most of who had not attended a mass before and during the pandemic.”
7.It is worth comparing the results of the study and the information on Catholics with studies conducted by the Institute for Catholic Church Statistics. See also: Gawroński, Sławomir, Dariusz Tworzydło, and Bajorek Kinga. 2021. Determinants for the Development of the Activity of the Catholic Church in Poland in the Field of Social Communication. Religions 12: 845.
The results of our study concerning students’ change in religious practices were discussed in the aspect of the suggested literature:
“However, in both groups of students there was a percentage of those who stopped religious practices during pandemic (8.6% of catholic school students versus 12.9% from public schools). This may also be discussed in the aspect of the report presented by GawroÅ„ski et al. 2021. Based on their fair literature analysis and interpretation, the authors found that in recent years persistent critical opinions towards the Catholic Church appeared. This suggests that the Church in Poland has been losing confidence in its image probably due to inefficient communication.” (GawroÅ„ski, S.; TworzydÅ‚o, D.; Bajorek, K. Determinants for the Development of the Activity of the Catholic Church in Poland in the Field of Social Communication. Religions 2021, 12, 845. https://doi.org/10.3390/rel12100845).
8.In lines 228-229 the authors write about Muslims and refer to a specific article on Turkey. Please justify how far this is justified.
Here, we cited the article including Muslims, as we wanted to discuss influence of different faiths not only catholic on the analysed aspects or to at least show that such researches are conducted in different religions as faith itself, regardless its name may be important factor influencing many aspects during pandemic.
9.Please explain in the research methodology what guided the selection of specific schools and to what extent the selection of the research group was purposeful.
As it was stated: “According to the aim of the study, to include religious adolescents we invited students from two kinds of high schools: public and Catholic.”
“Participants were recruited in four public high schools from Lodz and nearby City (Piotrków Trybunalski) (partner high schools of Medical University of Lodz) and four Catholic (non-public) schools from east Poland: Lodz, Ostróda, Legionowo, MiÅ„sk Mazowiecki) (Salesian School Complex).”
“The choice of the schools was related to certain public schools partnership with researchers’ University, which helped to cooperate with schools’ directors. Salesian School Complex is the biggest network of catholic schools and its structure involves division into 4 quadrants (regions), from which we asked east-central inspectorate to take part in the study as it involves Lodz and the surrounding cities.”
10.It is necessary to write on what basis the article draws the conclusions in lines 307 and 308.
This was clarified:
“We showed that in some adolescents’ opinions the pandemic caused family relations to be stronger, however this effect was modified by religiosity as more frequently attending believers less often reported such influence.”
- In the conclusion it is worth indicating what are the prospects for further research.
Added, as suggested:
“This observation should be verified in further studies, also in the aspect of health-related issues, e.g. knowledge and attitude towards prophylactic screening tests and prophylactic measures like obligatory and voluntary vaccinations. This may help to design and properly target appropriate educational programs.”
Reviewer 4 Report
The article analyzes very important research issues regarding faith, religion, family interactions and interest in health issues during the COVID-19 pandemic. The aim of the study is to verify whether the pandemic of COVID-19 changed religious practices and how religiosity moderated the influence of the pandemic on family interactions and attitudes towards health issues in adolescents. This study should be pursued.
One important change is necessary before publishing the article. Without some explanation, the study is partially incomprehensible to the vast majority of readers who do not know the specificity of the school system in Poland. Well, in a dozen or so sentences it is necessary to describe the division into state and non-state, public and non-public schools in Poland.
On the one hand, 85% of the population of Poland belongs to the Catholic Church when it comes to people baptized in this denomination. On the other hand, only a few percent of children and adolescents attend Catholic schools. In Poland, 70.000 students study in Catholic schools, including 44.000 primary school students (1.5 percent of the total number of primary school students in Poland) and 21.000 high school students (4.5 percent of the total number of students in this type of school).
This wider context shows that in many cases the choice of a Catholic school is a deliberate religious and moral decision of parents and children. The consequence of this situation is partly a specific axiological identity of students of Catholic schools in Poland.
Author Response
Dear Reviewer,
Thank you for your interest in our manuscript entitled " The influence of faith and religion on family interactions and interest in health issues during the COVID-19 pandemic - a study among Polish adolescents”. We would like to thank anonymous Reviewers for their detailed valuable comments and reviews, which helped to improve the manuscript. In this revision we addressed all your comments (in the text changes are marked in red colour). We hope that this revision meets with your approval.
Sincerely,
The Authors
Responses to Reviewer’s comments:
Reviewer 4
1.One important change is necessary before publishing the article. Without some explanation, the study is partially incomprehensible to the vast majority of readers who do not know the specificity of the school system in Poland. Well, in a dozen or so sentences it is necessary to describe the division into state and non-state, public and non-public schools in Poland.
On the one hand, 85% of the population of Poland belongs to the Catholic Church when it comes to people baptized in this denomination. On the other hand, only a few percent of children and adolescents attend Catholic schools. In Poland, 70.000 students study in Catholic schools, including 44.000 primary school students (1.5 percent of the total number of primary school students in Poland) and 21.000 high school students (4.5 percent of the total number of students in this type of school).
This wider context shows that in many cases the choice of a Catholic school is a deliberate religious and moral decision of parents and children. The consequence of this situation is partly a specific axiological identity of students of Catholic schools in Poland.
Thank you for your digression with some “ready- to-use” data – it was used in the paragraph describing the school system in Poland in the context of our study /as suggested, in the introduction/.
“The school system in Poland involves pre-school education (3-6 years), 8-year primary schools and secondary schools (4-year general, 5 year technical secondary and sectoral vocational schools) and higher education units. The schools may be public/state or non-public/none-state, which still may benefit from public funding (i.e. Salesian School Complex). They may be created by: legal bodies (associations, religious bodies, foundations, companies), and by individuals. Currently there are about 500 catholic schools in Poland. It is however interesting, as despite such high percentage of the population of Poland belongs to the Catholic Church, only a few percent of children and adolescents attend Catholic schools. In Poland, 70.000 students attend Catholic schools, including 21.000 high school students (4.5 percent of the total number of students in this type of school). This suggests that the choice of a Catholic school is a deliberate religious and moral decision of parents and children, which may suggest a specific axiological identity of students of Catholic schools in Poland. Moreover, most of catholic schools require so called catholic faith declaration (as in case of Salesian School Complex).”
Reviewer 5 Report
“Does faith and religion influence family interactions and interest in health issues during the COVID-19 pandemic? A study among Polish adolescents”
Review comments:
The introduction gives some basic background to the pandemic, but surely this is unnecessary as it is by now common knowledge the world over. Certainly readers of the journal will be very familiar with pandemic-related issues. The introduction does not make the parameters of the study very clear, however, only giving a broad reason as to its intent to examine religious practices vis-à-vis ‘family interactions and attitudes towards health issues’. I think that this would benefit from having the obvious background removed and more detail provided for the reader as to what the study is about and where it is situated within the research field (e.g. I could not understand what the sentences in Lines 50-52 is meant to communicate.).
Were the students given a chance to self-report on religious commitment? It seems to me that merely being a student at a Catholic school does not necessarily make one religious, nor that being a student at a public school would make one necessarily irreligious. Perhaps this was part of the questionnaire.
The questionnaire explanation section does indeed indicate that personal religiosity was asked about; this helps me follow the study better. The presumption though appears to solely be about Christian religiosity; I understand that within the cultural situation this is probably appropriate, but perhaps some kind of reasoning or explanation could benefit the reader.
What type of attitude was the question on encouraging preventive screening tests meant to reveal?
The paragraph on the results of expected/unexpected answers makes me wonder about the researchers’ assumptions prior to the study; perhaps these could be briefly outlined.
The summaries of other studies and related background material in the Discussion section would probably fit better at the beginning of the study, either as part of the Introduction or immediately following it. In my opinion, this would better place the questionnaire that was conducted within the wider research context. If so then the Discussion section could focus solely on analysis and commentary regarding the results of the questionnaire.
The few points of difference in the answers between regular mass attendees and frequent mass attendees were very interesting; more discussion about what factors may be at work in these instances could be called for if the authors agree. I’m also not sure that the assumption of religious values favoring the family means that practicing Catholic families would in general have better integration; this may be true, but I think it should be shown or phrased more carefully.
Author Response
Dear Reviewer,
Thank you for your interest in our manuscript entitled " The influence of faith and religion on family interactions and interest in health issues during the COVID-19 pandemic - a study among Polish adolescents”. We would like to thank anonymous Reviewers for their detailed valuable comments and reviews, which helped to improve the manuscript. In this revision we addressed all your comments (in the text changes are marked in red colour). We hope that this revision meets with your approval.
Sincerely,
The Authors
Responses to Reviewer’s comments:
Reviewer 5
1.The introduction gives some basic background to the pandemic, but surely this is unnecessary as it is by now common knowledge the world over. Certainly readers of the journal will be very familiar with pandemic-related issues. The introduction does not make the parameters of the study very clear, however, only giving a broad reason as to its intent to examine religious practices vis-à-vis ‘family interactions and attitudes towards health issues’. I think that this would benefit from having the obvious background removed and more detail provided for the reader as to what the study is about and where it is situated within the research field (e.g. I could not understand what the sentences in Lines 50-52 is meant to communicate.).
The summaries of other studies and related background material in the Discussion section would probably fit better at the beginning of the study, either as part of the Introduction or immediately following it. In my opinion, this would better place the questionnaire that was conducted within the wider research context. If so then the Discussion section could focus solely on analysis and commentary regarding the results of the questionnaire.
The manuscript was reorganized – some information from the discussion was moved to the introduction, but we kept some aspects discussed with the literature so the discussion remained “discussion”. Some basic and obvious information were removed from the introduction. The sentence in the highlighted lines was clarified”
“These statistics indicate that religiosity of adolescents and young adults are becoming weaker and there are more and more non-believers, however the current changes in the religiosity of Poles should be interpreted in the wider aspects and should combine different perspectives. As, on the other hand, the percentage of “strong believers” who practice several times a week remains stable or even increases. This indicates two different directions: secularization, atheization and anticlericalism, and contrary: stronger and a more conscious and consistent religiosity [18].”
2.Were the students given a chance to self-report on religious commitment? It seems to me that merely being a student at a Catholic school does not necessarily make one religious, nor that being a student at a public school would make one necessarily irreligious. Perhaps this was part of the questionnaire. The questionnaire explanation section does indeed indicate that personal religiosity was asked about; this helps me follow the study better. The presumption though appears to solely be about Christian religiosity; I understand that within the cultural situation this is probably appropriate, but perhaps some kind of reasoning or explanation could benefit the reader.
Yes, as you stated: being a student at a Catholic school does not necessarily make one religious, but in case of most Catholic schools catholic faith declaration is a formal requirement during recruitment process, as it was in Salesian School Complex included in our study. This makes our assumption that most of students who declared religiosity by attending religious practices were Catholics. This was clarified in the paragraph providing background of the school system in Poland:
“The school system in Poland involves pre-school education (3-6 years), 8-year primary schools and secondary schools (4-year general, 5 year technical secondary and sectoral vocational schools) and higher education units. The schools may be public/state or non-public/none-state, which still may benefit from public funding (i.e. Salesian School Complex). They may be created by: legal bodies (associations, religious bodies, foundations, companies), and by individuals. Currently there are about 500 catholic schools in Poland. It is however interesting, as despite such high percentage of the population of Poland belongs to the Catholic Church, only a few percent of children and adolescents attend Catholic schools. In Poland, 70.000 students attend Catholic schools, including 21.000 high school students (4.5 percent of the total number of students in this type of school). This suggests that the choice of a Catholic school is a deliberate religious and moral decision of parents and children, which may suggest a specific axiological identity of students of Catholic schools in Poland. Moreover, most of catholic schools require so called catholic faith declaration (as in case of Salesian School Complex).”
We did not included a specific question about self-reported religious commitment on purpose (we did not ask directly about the religiosity and the type of faith). Such question could be “not comfortable” for nowadays adolescents in Poland. The aim was to examine the influence of religiosity (not a specific religion) on different aspects. Of course, taking into consideration polish background probably all or almost all believers were Catholics (and for sure all from Salesian School Complex), but this was not a “must”. And as a measure of their commitment we accepted ‘attendance in religious practices’ before pandemic.
3.What type of attitude was the question on encouraging preventive screening tests meant to reveal?
The idea behind this question involved examining student’s attitude to health issues in respondent’s family and their care about close relatives. Screening teste are important aspect of the health self-concern. In fact in Poland there is a problem of attendance in prophylactic programmes (it is low…), and students have educational programmes concerning these issues so they have appropriate knowledge to influence/encourage their relatives to take part in these programmes (if they care about their health).
4.The paragraph on the results of expected/unexpected answers makes me wonder about the researchers’ assumptions prior to the study; perhaps these could be briefly outlined.
Such description is connected with reporting residues, which is explained in section statistical analysis (and was further clarified):
“In order to assess which responses influenced the results, we calculated the residues, that is, standardized differences between the actual number of given responses and their theoretical number, for instance, what it would be if there was no relationship between the analyzed variables. Cases where the number of responses was greater/less than the theoretical number were marked in table 2, 3 and 4 (answers given more/less often than predicted in relation to their theoretical number).”
5.The few points of difference in the answers between regular mass attendees and frequent mass attendees were very interesting; more discussion about what factors may be at work in these instances could be called for if the authors agree. I’m also not sure that the assumption of religious values favouring the family means that practicing Catholic families would in general have better integration; this may be true, but I think it should be shown or phrased more carefully.
We think that the differences you highlighted are worth further studies and explanations!
“It would be worth to further examine the differences in attitude to different health and social aspects in relation to the “strength” of faith and search for the possible explanations of these differences.”
This “unfortunate” sentence was changed:
“It may be due to the fact that religious families, usually favouring the family, could have had better integration before the pandemic, which results from the priority of the family in the religious value system.” so now we refer to the religious families /regardless the “type” of religion/.
Round 2
Reviewer 3 Report
I accept the article as amended and supplemented.